# Performance of COVID-19 case-based surveillance system in FCT, Nigeria, March 2020 –January 2021

**Chikodi Modesta Umeozuru**[1,2]*, **Aishat Bukola Usman**[2], **Abdulhakeem Abayomi Olorukooba**[3], **Idris Nasir Abdullahi**[4‡], **Doris Japhet John**[5‡], **Lukman Ademola Lawal**[5‡], **Charles Chukwudi Uwazie**[1,2‡], **Muhammad Shakir Balogun**[2‡]

**1** Nigeria Field Epidemiology and Laboratory Training Program, Abuja, Nigeria, **2** African Field Epidemiology Network (AFENET), Abuja, Nigeria, **3** Department of Community Medicine, Ahmadu Bello University, Zaria, Kaduna, Nigeria, **4** Department of Medical Laboratory Science, Ahmadu Bello University, Zaria, Kaduna State, Nigeria, **5** Department of Public Health, Federal Capital Territory Administration, Abuja, Nigeria

☯ These authors contributed equally to this work.
‡ INA, DJJ, LAL, CCU, and MSB also contributed equally to this work.
* chikodi.umeozuru@gmail.com

**Data Availability Statement:** Data cannot be shared publicly because of third party confidentiality issue. Data will be available on request with the approval of the Federal Capital

## Abstract

### Introduction

The emergence of novel SARS-CoV-2 has caused a pandemic of Coronavirus Disease 19 (COVID-19) which has spread exponentially worldwide. A robust surveillance system is essential for correct estimation of the disease burden and containment of the pandemic. We evaluated the performance of COVID-19 case-based surveillance system in FCT, Nigeria and assessed its key attributes.

### Methods

We used a cross-sectional study design, comprising a survey, key informant interview, record review and secondary data analysis. A self-administered, semi-structured questionnaire was administered to key stakeholders to assess the attributes and process of operation of the surveillance system using CDC's Updated Guidelines for Evaluation of Public Health Surveillance System 2001. Data collected alongside surveillance data from March 2020 to January 2021 were analyzed and summarized using descriptive statistics.

### Results

Out of 69,338 suspected cases, 12,595 tested positive with RT-PCR with a positive predictive value (PPV) of 18%. Healthcare workers were identified as high-risk group with a prevalence of 23.5%. About 82% respondents perceived the system to be simple, 85.5% posited that the system was flexible and easily accommodates changes, 71.4% reported that the system was acceptable and expressed willingness to continue participation. Representativeness of the system was 93%, stability 40%, data quality 56.2% and timeliness 45.5%, estimated result turnaround time (TAT) was suboptimal.

Territory Health Research Ethics Committee, for researchers who meet the criteria for access to confidential data. A non-author point of contact for Federal Capital Territory Administration: The Secretary of Health and Human Services Secretariat, Federal Capital Territory Administration, Abuja, Nigeria. Dr. Abubukar Tafida (drtafida2@gmail.com).

**Funding:** The author(s) received no specific funding for this work.

**Competing interests:** The authors have declared that no competing interests exist.

## Conclusion

The system was found to be useful, simple, flexible, sensitive, acceptable, with good representativeness but the stability, data quality and timeliness was poor. The system meets initial surveillance objectives but rapid expansion of sample collection and testing sites, improvement of TAT, sustainable funding, improvement of electronic database, continuous provision of logistics, supplies and additional trainings are needed to address identified weaknesses, optimize the system performance and meet increasing need of case detection in the wake of rapidly spreading pandemic. More risk-group persons should be tested to improve surveillance effectiveness.

## Introduction

Since its emergence in December 2019, the novel coronavirus SARS-CoV-2 has caused a pandemic of Coronavirus Disease 19 (COVID-19) which has spread exponentially all over the world. As of July 2021, the disease had affected 220 countries with over 190 million confirmed cases and 4 million deaths worldwide [1]. Although the case fatality rate of COVID-19 which is estimated at 2%–3% is lower than that of MERS (40%) and SARS (10%), the pandemic associated with COVID-19 has been more severe [2]. The first COVID-19 case in Nigeria was recorded on 27th February [3], this index case brought about the activation of COVID-19 Emergency Operation Center (EOC) at national and sub-national levels. COVID-19 surveillance in Nigeria from 27th February to 31st January 2021 recorded a total of 131,242 confirmed cases; 104,989 recoveries, 1,586 deaths with case fatality rate (CFR) of 1.2% and 24,667 active cases [4]. The Federal Capital Territory (FCT) reported its first COVID-19 case on 20th March 2020 following laboratory confirmation of the first three COVID-19 cases. The FCT has since become the second epicenter of COVID-19 in Nigeria after Lagos State with a total of 16,863 confirmed cases from 20th March to 31st January 2021: 10,983 recoveries, 126 deaths (CFR 1.1%) and 5,754 active cases. The highest proportion of COVID-19 cases and deaths was seen among age groups 31–40 years and 61–70 years, respectively; and males accounted for a higher proportion of confirmed cases and deaths [4, 5].

Case detection and contact identification remain the key surveillance objectives for effective containment of COVID-19. A robust surveillance system is essential for correct estimation of the burden of the disease and containment of the pandemic [6]. To adequately measure the level of COVID-19 pandemic containment, there is need for a robust local and regional epidemiological data [7]. COVID-19 surveillance aims to enable public health authorities to reduce transmission of COVID-19 in the state, thereby limiting associated morbidity and mortality [8]. COVID-19 is captured as a mandatory notifiable case-based disease under the Integrated Disease Surveillance and Response (IDSR) 001 with requirements for immediate reporting. FCT Public Health Department (PHD) coordinates COVID-19 surveillance in FCT and is responsible for the review of data generated in the state, from which information is used for immediate public health action. Surveillance information flows from the lower to higher levels for onward public health actions based on the final laboratory outcome, while feedback goes in reverse direction with all reporting levels adequately captured in the system Fig 1. COVID-19 surveillance utilizes both active and passive surveillance.

Specific objectives of COVID-19 surveillance system are: to enable rapid case detection, conduct disease control interventions, identify, follow-up and quarantine contacts, detect and

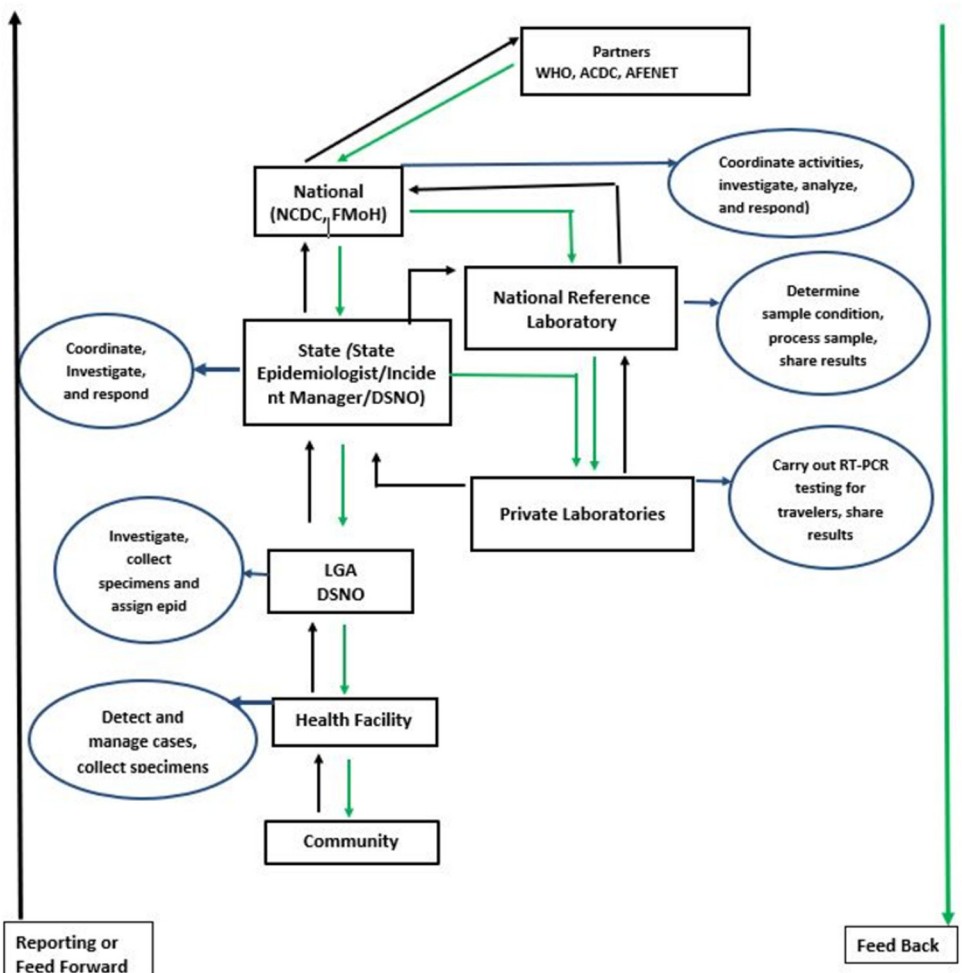

**Fig 1. Information flow in FCT COVID-19 surveillance system.**

contain clusters especially among vulnerable and high-risk populations, guide the implementation and adjustment of targeted control measures, monitor and predict the impact on healthcare systems and monitor longer-term epidemiologic trends and evolution of SARS-CoV-2 virus [8, 9].

The system is mainly funded by the Federal Capital Territory Administration (FCTA). The Federal Government of Nigeria through the Nigeria Centre for Disease Control (NCDC) provides trainings, funds the laboratory which is a fundamental aspect of COVID-19 surveillance and provides updated guidelines. Partners such as World Health Organization (WHO), the World Bank through Regional Disease Surveillance Systems Enhancement (REDISSE) and Africa Centre for Disease Control (ACDC) also provide funding and technical support to the system.

The FCT COVID-19 surveillance system like every other public health system is prone to challenges and setback. Although the system has been established and operational since the inception of the pandemic, there is limited information about the overall performance. Evaluation of the COVID-19 surveillance system will ascertain if the system is meeting the set objectives and whether the attributes are effective enough to accomplish these objectives [10, 11]. Furthermore, information from the evaluation is crucial for improving the system as well as

enhancing preparedness for possible future outbreaks of COVID-19 and other emerging infectious diseases. Several attributes of a surveillance system impact its capacity to monitor health events efficiently [12]. They include usefulness, simplicity, flexibility, data quality, acceptability, sensitivity, positive predictive value, representativeness, timeliness, and stability. The evaluation of public health surveillance systems entails an assessment of these nine system attributes [13–15]. We evaluated the performance of the COVID-19 case-based surveillance system in FCT, Nigeria and assessed its key attributes.

## Methods

### Study setting

The FCT which is in North-central Nigeria has a landmass of approximately 7,315 km$^2$ and it is situated within the savannah region with moderate climatic conditions. The territory is currently made up of six local government areas (LGAs) also called area councils, namely: Abaji, Abuja Municipal, Bwari, Gwagwalada, Kuje and Kwali [16]. The FCT has 2016 projected population of 3,564,126 [17]. There are 19 NCDC accredited Reverse Transcription Polymerase Chain Reaction (RT-PCR) testing laboratories for COVID-19: three government owned laboratories (National Reference Laboratory, Defense Reference Laboratory and FCTA PCR Laboratory) and 16 private laboratories that cater for persons of interest (POI). There are four active COVID-19 treatments centres in FCT: Gwagwalada Specialist Hospital, National Hospital Abuja, This Day Dome and Idu treatment center. There are six active sample collection centres: Maitama District Hospital, Nyanya General Hospital, This Day Dome, Gwagwalada Specialist Hospital, FCT Public Health Department, Federal Medical Center Jabi and Kubwa General Hospital.

The surveillance system uses a digital surveillance tool called Surveillance Outbreak Response Management and Analysis System (SORMAS) to report COVID-19 cases. SORMAS is an open-source real-time mobile and web application used as an electronic health surveillance database. The NCDC adopted SORMAS as its primary digital surveillance platform for implementing the IDSR system and customized it for the surveillance of priority diseases of public health importance in Nigeria, a COVID-19 module was developed and added to SORMAS in January 2020 [3]. The essential surveillance data for COVID-19 is reported, compiled, and analyzed daily, with zero reporting when there are no cases. Data is usually compiled at the state and national levels, more in-depth analysis on age, gender, testing patterns, comorbidities and risk factors, symptomatology and severity, etc. are also conducted periodically.

Other existing surveillance approaches are used along with the essential elements of comprehensive surveillance for COVID-19, these include participatory surveillance which enables members of the public to self-report signs or symptoms, this relies on voluntary reporting and is frequently facilitated by dedicated smartphone applications and telephone hotlines which were made available to the public for advice and referral to health care service, this provides an early indication of disease spread in the community [10]. The data generated are securely stored electronically and in hard copies at the FCT COVID-19 EOC. Data are not disclosed to any party unless the purpose of utilization is clarified and fully authorized by FCT ethical committee. The data is also electronically backed up and access is limited to authorized key persons. The FCT COVID-19 surveillance report captures data on suspected COVID-19 cases and deaths, active case search, community reporting via the COVID-19 toll-free lines, case investigations, health facility (deaths and suspected cases), laboratory data and treatment center admissions, outcomes and discharges.

## Study design

An evaluation of the COVID-19 surveillance system was conducted using a cross-sectional study design. The surveillance system was evaluated using the "2001 United States Centers for Disease Control's updated guidelines for Evaluating Public Health Surveillance Systems".

## Study population

A total of 30 stakeholders involved in the COVID-19 surveillance system in FCT were interviewed, these includes: The Director Public Health, State Epidemiologist, Assistant State Epidemiologist, Incident Manager, Laboratory pillar lead, State Disease Surveillance and Notification Officer (DSNO), assistant state DSNO, local government area DSNOs, SORMAS state surveillance officer, case managers, Laboratory Scientists, community health workers and Partners (WHO, ACDC).

## COVID-19 case definitions

**Suspected case.** "A suspect case is defined as any person (including severely ill patients) presenting with fever, cough or difficulty in breathing AND who within 14 days before the onset of illness had any of the following exposures: History of travel to any country with confirmed and ongoing community transmission of SARS-CoV-2 **OR** Close contact with a confirmed case of COVID-19 **OR** Exposure to a healthcare facility where COVID-19 case(s) have been reported" [18].

**Confirmed case.** "A person with laboratory confirmation of SARS-CoV-2 infection with or without signs and symptoms" [18].

**Probable case.** "A probable case is defined as a person who meets the criteria for a suspect case AND for whom testing for COVID-19 is inconclusive or for whom testing was positive on a pan-coronavirus assay" [18].

**Contact case.** "someone who had contact (within 1 meter) with a confirmed case during their symptomatic period, including one day before symptom onset" [18].

## Data collection

A mixed data collection approach was adopted, this includes key informant interviews (KII), surveys, record reviews and analysis of FCT COVID-19 surveillance data from March 2020 to January 2021. Data collection was done in February 2021. The KII was conducted with four key stakeholders (state epidemiologist, incident manager, state DSNO and state SORMAS Surveillance officer). Purposive sampling method was used to select the four key stakeholders for KII. Survey was conducted using a self-administered semi-structured questionnaire which was administered to 30 stakeholders to obtain their inputs in describing the system and assess their perception of attributes of the surveillance system. The questionnaire had two sections: Section A collected sociodemographic information of respondents while Section B was on the surveillance system attributes: simplicity, flexibility, data quality, acceptability, sensitivity, positive predictive value, representativeness, timeliness, and stability. The questionnaire outlined various indicators (range 2–5) for assessment of each system attribute. The questionnaire and key informant interview guide were adapted from the United States Centres for Disease Control and Prevention (CDC), 2001 Updated Guidelines for Evaluation of Public Health Surveillance Systems [14]. SORMAS was the primary data source for this study, FCT COVID-19 surveillance data comprising of epidemiological and laboratory variables on suspected and confirmed cases between March 2020 and January 2021 were extracted from the SORMAS platform on excel datasheet. Variables of interest were socio-demographic characteristics, date case was

reported, laboratory result and outcome of illness. The surveillance data was used to assess the usefulness of the system. Documents relevant to COVID-19 surveillance were reviewed; these includes Form A0: case investigation form (CIF), Form A1: case initial reporting form for confirmed cases (Day 1), Form A2: case follow-up reporting form (Day 14–21), Form B1: contact initial reporting form for close contacts (Day 1), Form B2: contact follow-up reporting form for close contacts (Day 14–21), National Technical Guidelines for IDSR 2019 version, National Interim Guidelines for Clinical Management of COVID-19 version 4, the First Few X (FFX) cases and contact investigation protocol for 2019-novel coronavirus (2019-nCoV) infection.

## Data analysis

We analyzed the extracted surveillance data using Epi Info 7.0 and Microsoft Excel 2016. Data output was summarized into descriptive statistics (frequencies and proportions) using charts and tables. Case fatality rate was calculated by dividing the number of deaths from COVID-19 over the time period by the number of individuals that tested positive to COVID-19 during that time, the resulting ratio was then multiplied by 100 to give a percentage. Positivity rate was calculated by dividing the number of people that tested positive by the number of people who were tested. Attack rate was calculated by dividing the total number of cases during the time period by the total population at risk. Content analysis procedure was employed for data collected from the KIIs. We analyzed the data from survey questionnaires and scored the responses for various system attributes. To achieve consistency and comparability of findings, we used the evaluation method and scoring system utilized for influenza surveillance evaluations conducted in other African countries [11, 19]. A scale from 1 to 3 was used to provide a score for each indicator as follows: < 60% scored 1 (poor performance); 60–79% scored 2 (moderate performance); ≥80% scored 3 (good performance [19]. The scores allotted to each indicator were averaged for all indicators evaluated within each attribute to give an overall score for a particular attribute. The nine evaluated attributes were then averaged to get an overall score for the surveillance system.

## Ethical consideration

Ethical approval with approval number: FHREC/2021/01/18/26-03-21 was obtained from the FCT Health Research Ethics Committee, written informed consent was also obtained from stakeholders.

## Results

### Findings from KII with stakeholders

Some challenges affecting the surveillance system as indicated by the key stakeholders include non-harmonization of treatment protocol amongst the treatment centres, prolonged result turnaround time, incomplete reporting, inadequate funding, sub-optimal testing in all area councils except Abuja municipal, low case-contact ratio due to prolonged result turnaround time (TAT) and insufficient number of personnel for contact tracing and inadequate office space for the EOC thereby not encouraging adequate physical distancing. The stakeholders also reported community stigmatization and denial as a major challenge, leading to low testing, poor disclosure, inadequate contact tracing and poor adherence to non-pharmaceutical interventions (NPI) by FCT residents due to low-risk perception.

Other reported challenges include inadequate vehicles for response activities, increasing number of healthcare workers' infection, cumbersome process of phone number extraction from SORMAS leading to delay in result dissemination in addition to limited number of

personnel to carry out this task, low/stock out of some response commodities, discordant results from private laboratories and inadequate holding area for suspected cases. Stakeholders stated that the SORMAS platform is extremely slow and so hinders real-time data capture on the field. As a result, a huge disparity was always seen between data captured on SORMAS and the actual surveillance data. Tablets and SIM cards are available for SORMAS operation but there is no designated fund to carter for data and airtime, as a result, some personnel often do out of pocket spending to keep the platform updated.

## FCT COVID-19 surveillance system attributes

**Usefulness.** COVID-19 surveillance as part of IDSR was established to provide prompt detection and response to COVID-19 pandemic. A total number of 69,338 suspected cases of COVID-19 were reported in FCT between March 2020 to January 2021, 12,595 (18%) samples tested positive with RT-PCR, Table 1.

A total of 125 deaths were recorded among confirmed cases giving an overall CFR of 1%. The system showed distribution of these cases across the six local governments areas of the state, Abuja municipal recorded the highest attack rate (56.0) followed by Gwagwalada (15.1) with an overall attack rate of 35.1. Out of 2,384 suspected cases among healthcare workers, 604 tested positive for COVID-19, which translates into a prevalence of 25.3% among healthcare workers (HCWs). Prevalence was 18.5% among traders, 16.2% among students, and 14.1% among civil servants. The proportion of HCW infection was highest among nurses (28.8%) and doctors (25.8%). Distribution of confirmed cases showed a propagated pattern as seen in Fig 2, with peaks seen in week 52 and 51, a sharp decline in number of cases was observed from week 1 in 2021.

Case fatalities were recorded across the entire period with highest number of deaths seen in weeks 28 and 53. About 74% of COVID-19 cases admitted at the treatment centers made full recovery, 1% died while outcome was unknown for 25%. The system successfully detected the second wave of the outbreak which started from week 48 and accounted for more than half of the COVID-19 cases recorded in the entire period Fig 3. System usefulness had a total score of 98%.

**Simplicity.** The FCT COVID-19 Surveillance System has a simple structure, the structure of the system is such that data flow occurs uninterrupted from communities to the national level as indicated in Fig 1. All respondents reported that data tools were easy to use and the COVID-19 case definitions were easy to understand. About 95% reported that COVID-19 posters and other Information, Education and Communication (IEC) materials were displayed in their facilities. Out of those that reported availability of COVID-19 IEC materials in their facilities, 95% stated the IEC materials contained COVID-19 case definitions while 53% said they contained COVID-19 case management guidelines. About 76% claimed they report all

**Table 1. Distribution of confirmed COVID-19 cases in FCT by LGA from March 2020 –January 2021.**

| LGA | Suspected | Confirmed | Deaths | CFR | Positivity rate | Population size [20] | Attack rate/10,000 population |
|---|---|---|---|---|---|---|---|
| Abaji | 649 | 47 | 0 | 0% | 7% | 146, 600 | 3.2 |
| Abuja Municipal | 60641 | 11018 | 110 | 1% | 18% | 1, 967, 500 | 56.0 |
| Bwari | 4419 | 730 | 7 | 1% | 17% | 581, 100 | 12.6 |
| Gwagwalada | 2554 | 605 | 7 | 1% | 24% | 402, 000 | 15.1 |
| Kuje | 886 | 163 | 1 | 1% | 18% | 246, 400 | 6.6 |
| Kwali | 189 | 32 | 0 | 0% | 17% | 218, 400 | 1.5 |
| Total | 69338 | 12595 | 125 | 1% | 18% | 3, 564, 000 | 35.3 |

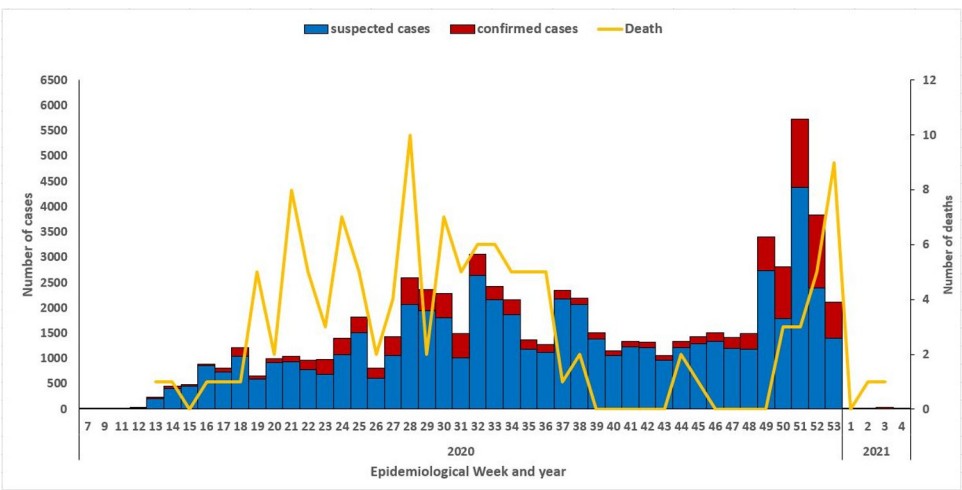

**Fig 2. Distribution of suspected and confirmed COVID-19 cases in FCT from March 2020-January 2021.**

suspected cases of COVID-19, 48%, 19% and 5% report suspected cases to the next level through phone calls, 19% report on CIF and 5% report through SORMAS. Eighty-one percent reported that the system constantly detects an increase in COVID-19 cases. Twenty-four percent of respondents reported detected cases to other organizations and 14% of the respondents who send case reports to other organizations use other forms apart from the COVID-19 surveillance forms to send the reports. The estimated time spent on collection, entering, editing, analyzing, storing, backing up and transfer of data ranged from 25 minutes to 72 hours among respondents. About 71% of respondents reported ease of working within the system in terms of operation: workload, workflow, the flow of information and inter-unit relationship. About 81% of respondents reported having adequate number of staff for data collection and those

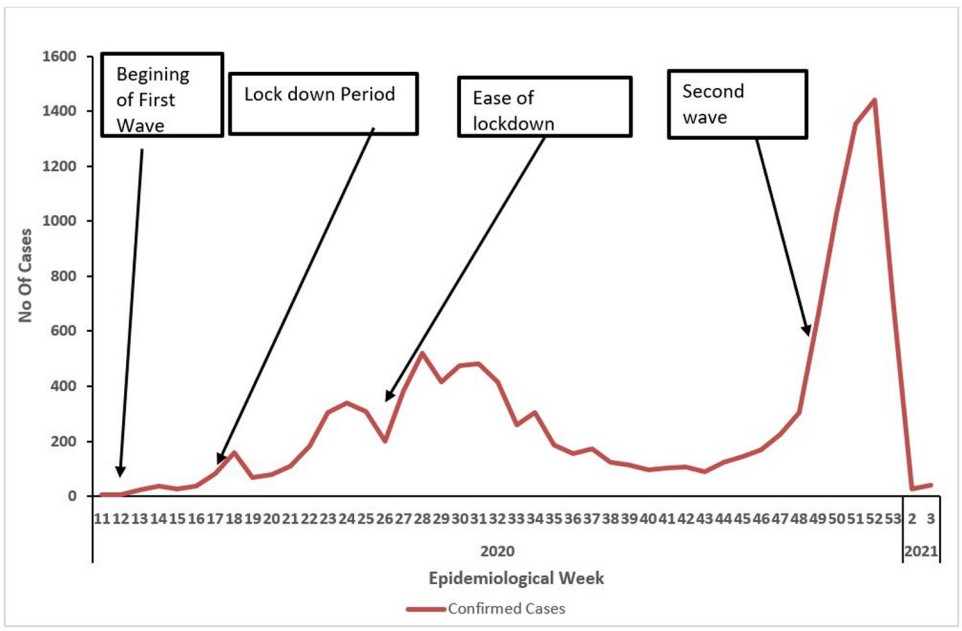

**Fig 3. Trend of COVID-19 confirmed cases in FCT from March 2020—January 2021.**

who reported inadequate number of staff for data collection needed an average of 13 staff for optimal task performance. The overall score for system simplicity was 81.5%.

**Flexibility.** Ninety-one percent of respondents reported that changes in the surveillance system were easily accommodated by the data collection tools and 81% stated that new data elements were easily reported as part of the monthly report. The overall score for system flexibility was 85.5%

**Acceptability.** All respondents expressed willingness to continue their participation in the surveillance system, however, 71% had challenges affecting work efficiency. About 86% had made suggestions/comments about improving the system, the suggestions made include: engagement of more support staff 28%, financial support for staff 28%, improvement of result turnaround time (TAT) 22%, transport logistic support 22%, improvement of electronic database 17%, additional training 17%, automation of result dissemination 17%, improvement of surveillance funding 17%, provision of adequate work tools 17%, more spacious work space 17%, timely supply of essential materials 11%, simplification of CIFs for suspected cases 6%, collaboration with other agencies 6% and proper supervision at sample collection sites 6%. About 57% of those that made suggestions reported that their suggestions were considered. About 90% feel the system appreciates them for doing their job. All respondents needed additional support to carry out their tasks effectively. The overall system acceptability score was 71.4%.

**Timeliness.** About 90% of respondents reported availability of written policies on the timeliness of data reporting, 67% reported having challenges with sending reports on a timely basis. The challenges reported includes delays in receiving results from the laboratories, problems with data harmonization, lack of funds for transmitting reports, overwhelming nature of workload, inadequate manpower, delay in reporting from lower levels, lack of/poor internet access and poor transport logistics. About 71% incurred additional costs for reporting, ranging from 3000 naira to 100,000 Naira monthly. Respondents reported that TAT ranged from 24 hours to two weeks, the optimal TAT being 48 hours, TAT was suboptimal as only 29% reported TAT of 24–48 hours. About 76% of respondents reported data on daily basis, 19% reported weekly while 5% reported monthly. The timeliness and completeness of reporting were 63% and 70% respectively which were below the 80% and 100% targets respectively, Fig 4. The overall score for timeliness was 45.5%.

**Sensitivity.** About 97% of respondents were satisfied with the COVID-19 case definitions, 85% felt the system could detect all cases of COVID-19, 10% reported occurrence of false-negative results and 5% reported delay in the release of patients' results. About 95% of respondents said the system can correctly detect new cases of COVID-19. Nineteen percent reported frequent cases of misdiagnosis, the problems listed includes: misunderstanding of COVID-19 signs and symptoms with other infectious diseases, positive cases that turn out negative after a retest within 24 hours and lack of community testing. Respondents made the following suggestions for improvement of case detection: siting of sample collection centres all LGAs, improvement of TAT, raising the index of suspicion among healthcare workers, increased awareness in local languages, establishment of new testing sites, improvement of sensitivity of case definition, prompt release of result by testing laboratories, more trainings and improvement of sample collection method. A review of FCT COVID-19 surveillance data from March 2020 to January 2021 showed the system was able to detect 12,595 confirmed cases out of a total of 69,338 suspected cases within the period under review. The overall score for system sensitivity was 90.6%.

**Positive predictive value.** The FCT COVID-19 surveillance system detected a total of 69,338 suspected cases from March 2020—Jan. 2021. Samples were collected from these cases and tested at the laboratory. Out of these, 12,595 tested positive by RT-PCR

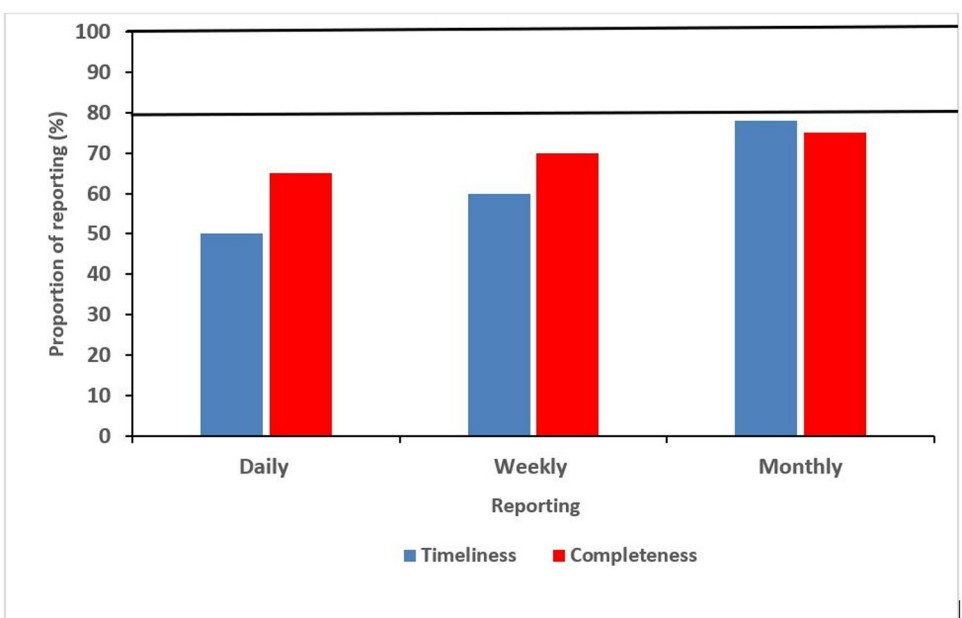

**Fig 4. Time taken to complete reports in FCT COVID-19 surveillance system, March 2020-January 2021.**

Therefore, the positive predictive value for FCT COVID-19 surveillance system = (12,595/ 69,338) × 100 = 18%.

**Data quality.** About 76% of respondents described the level of completeness of data generated from the COVID-19 surveillance system as partially complete while 57% reported that the data generated was partially valid. About 71% reported having been supervised on data management with an average of three supervisions in the last six months, more than 62% received feedback upon completion of supervision. About 5% of respondents rated the care exercised in completing surveillance forms and data management as excellent, 19% rated very good, 52% good, 19% fair and 5% poor. About 49% of respondents stated that feedback from SORMAS was inadequate, the challenges encountered include poor TAT, incomplete CIFs, most results not updated on SORMAS and problems with using SORMAS on a real-time basis for data capture. The overall score for the data quality of the system was 56.2%.

**Representativeness.** A review of the state SORMAS COVID-19 line-lists from March 2020 to January 2021 showed the FCT COVID-19 surveillance system captures people of all ages Fig 5.

Distribution of suspected and confirmed cases was seen across the six LGAs in FCT during the period under review. All respondents said the system captures people of all ages and 86% reported the system captures people from all geographical locations of FCT. Age groups 30–34, 35–39 and 25–29 years were mostly affected constituting about 14%, 13.9% and 13.8% of the confirmed cases respectively, Table 2, males accounted for a higher proportion of cases (56%). The representativeness score was 93%. CFR was highest among age group 70–79 and 80–89 years, Table 2.

**Stability.** About 95% of respondents received feedbacks from the next level. About 67% used the data collected to make informed decisions, planning of response and for analysis for transmission to policymakers in the state, while 33% reported they simply transmitted to the next level. About 67% claimed the system has been interrupted in the past due to inadequate funding, 52% claimed it was interrupted due to inadequate staff, 33% reported interruption

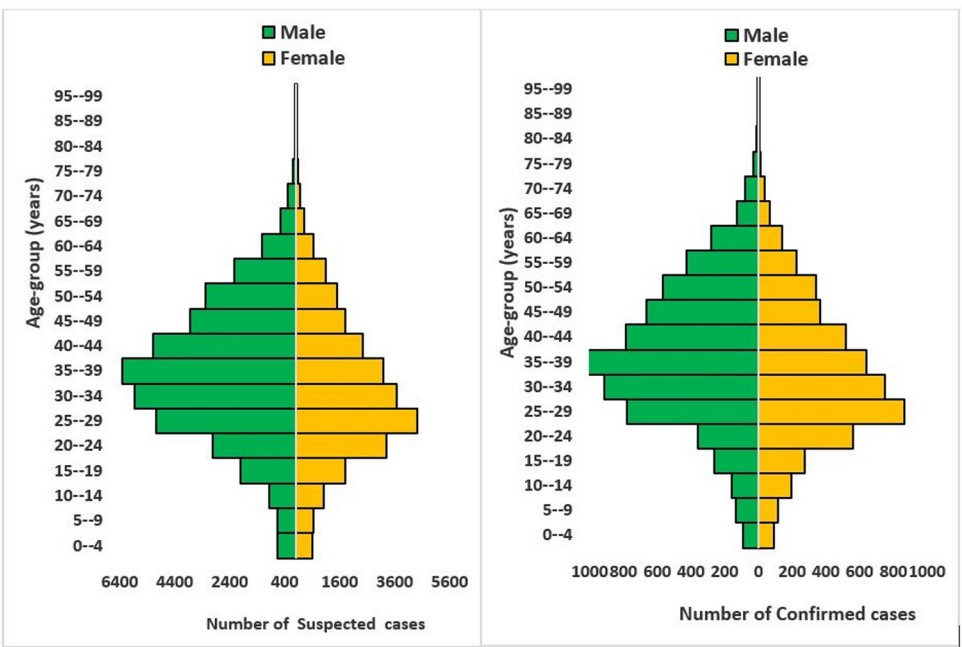

**Fig 5. Age-sex distribution of COVID-19 cases in FCT from March 2020-January 2021.**

due to stock out of consumables, 5% due to HCW infection and 5% due to strike action. Only 48% had stipends to carry out their tasks. About 76% needed more resource to carry out their duties effectively: funding (38%), human resource (34%) and material resources (28%), the material resources needed include internet services, adequate PPEs, data tools and transport logistics. Overall system stability score was 40%.

Having assessed all key system attributes, system usefulness scored the highest followed by flexibility while stability had the lowest score, Fig 6. The system had an average overall score of 73.5%.

## Discussion

The main goal of surveillance during outbreak management is to detect cases early in order to mount effective public health action to reduce the transmission. The FCT COVID-19

**Table 2. Age-sex distribution of FCT COVID-19 confirmed cases and deaths, March 2020 -January 2021.**

| Age Group | Female | | | Male | | | Total | | |
|---|---|---|---|---|---|---|---|---|---|
| | Confirmed | Death | CFR (%) | Confirmed | Death | CFR (%) | Confirmed | Death | CFR (%) |
| 0–9 | 207 | 0 | 0 | 225 | 1 | 0.4 | 432 | 1 | 0.2 |
| 10–19 | 469 | 0 | 0 | 420 | 0 | 0 | 889 | 0 | 0 |
| 20–29 | 1425 | 1 | 0.07 | 1140 | 1 | 0.09 | 2565 | 2 | 0.08 |
| 30–39 | 1388 | 2 | 0.1 | 1933 | 3 | 0.2 | 3321 | 5 | 0.15 |
| 40–49 | 885 | 4 | 0.5 | 1449 | 14 | 1 | 2334 | 18 | 0.8 |
| 50–59 | 562 | 9 | 1.6 | 990 | 25 | 2.5 | 1552 | 24 | 1.5 |
| 60–69 | 208 | 6 | 2.9 | 408 | 19 | 4.7 | 616 | 25 | 4.1 |
| 70–79 | 46 | 1 | 2.2 | 107 | 11 | 10.3 | 153 | 12 | 7.8 |
| 80–89 | 15 | 1 | 6.7 | 12 | 1 | 8.3 | 27 | 2 | 7.4 |
| 90–99 | 3 | 0 | 0% | 1 | 0 | 0 | 4 | 0 | 0% |
| Total | 5208 | 24 | 0.46 | 6685 | 75 | 1.1 | 11893 | 99 | 1 |

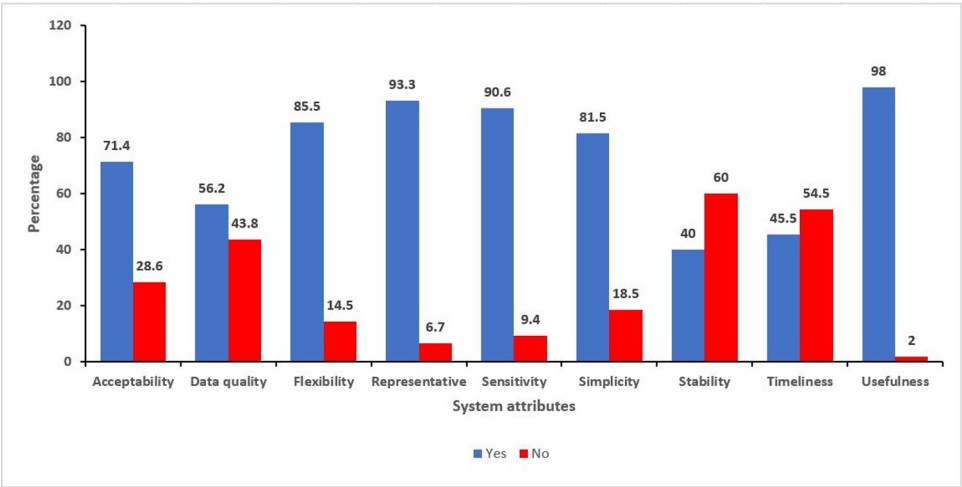

**Fig 6. Distribution of respondents' perception on FCT COVID-19 surveillance system attributes.**

surveillance system exhibits some of the attributes of a good surveillance system, however, poor stability, data quality and timeliness were limitations. Our findings showed that the surveillance system had an overall moderate performance, this implies that the system was not performing optimally although the system meets the initial objectives of the surveillance. This corresponds with the findings in similar studies conducted on Malaria Surveillance system in Kano State [10] and Avian Influenza Surveillance System in Enugu State, Nigeria, [21], where poor data quality, instability and non-representativeness of surveillance reports were the major limitations. We found that FCT COVID-19 surveillance system was effective in detecting cases in the communities. This was possible through periodic community testing and targeting suspected risk groups and persons of interest (POI), especially people coming into the state from countries with high burden. HCWs were identified as a high-risk group due to the high positivity rate recorded among them. The highest proportion of HCW infections occurred among nurses and doctors, this is worrisome considering that they usually have the first contact with patients. Esohe et al made a similar observation in a study of facility-based surveillance activities for COVID-19 infection and outcomes among healthcare workers in a Nigerian tertiary hospital [22]. Targeting risk groups for testing can improve the effectiveness of COVID-19 surveillance in settings where mass testing is not feasible. FCT COVID-19 surveillance system was found to be useful because it detects the disease in a way that permits early treatment, prevention and control. The system also provides estimates of COVID-19 related morbidity and mortality, including the identification of factors associated with the disease and detects trends that signal changes in the incidence of the disease. The system provided measurement of deaths attributed to COVID-19, which is a key indicator of the overall impact of the pandemic. Data generated from the system permits assessment of the effect of prevention and control programs and leads to improved clinical, behavioral, social, policy and environmental practices.

The system displayed good simplicity and ease of usage as data flow across the system occurs seamlessly and uninterruptedly and the data tools were deemed easy to use. The simplicity of this system could be attributed to the observed ease of application of the case definitions thereby making it easy to ascertain a case of COVID-19. However, our findings show that one of the major challenges facing the fight against COVID-19 is the similarities between its symptoms with many other popular diseases. Similarly, in an evaluation of health

surveillance systems in South Africa, about three-quarters of healthcare providers acknowledged that the operations of the existing surveillance system were simple and understandable [23]. The COVID-19 surveillance system was considered flexible since it was well integrated into IDSR, with paper-based forms and electronic forms (SORMAS) used for real-time data collection and reporting. The system was able to adapt changes according to needs and operational demands with minimal additional costs, this finding was in line with that of a similar study in Pakistan by Atifa et al. [6]. Although simplicity and flexibility were found to be major strength of the system, the system requires steady considerable amount of funding for smooth uninterrupted operation.

The system had good acceptability at the state and LGA levels, although more than half of the respondents had challenges affecting work efficiency and all respondents needed extra support for effective operation. Effort needs to be made by the state government and partners to address these challenges and provide the needed support to stakeholders in the system. Despite the reported challenges, all respondents were willing to continue participating in the system. About two-third of respondents perceived their contributions and suggestions within the system as being valued. In contrast, elsewhere, more than half of the health workers were dissatisfied with their involvement in surveillance activities [11]. Similarly, less than half of the respondents in a study in South Africa were willing to be involved in activities within the surveillance system [23]. Additional findings showed that majority of respondents recognized COVID-19 to be of high priority compared to other diseases, thus they were willing to engage in surveillance activities. Increased health worker participation in routine surveillance reporting is dependent on the perceived simplicity of the system [24]. Therefore, the high acceptability of the system can be attributed to simplified data tools, case definition and training materials which ease understanding. The overall timeliness of the system was poor, although there were written policies on the timeliness of data reporting, challenges were encountered that affected timely reporting and prolonged TAT from the testing laboratory was a major setback as this delayed effective public health action and timely containment. These finding aligns with a similar study that evaluated the timeliness and completeness of laboratory-based surveillance of COVID-19 cases in England which reported delays in timeliness, the delays occurring mostly in the first stage of the reporting process, before laboratory information is keyed onto the surveillance platform [25]. Data-driven insights to guide public health decision-making for the pandemic response rely majorly on complete and timely data on laboratory-confirmed cases. SORMAS platform which is the primary data source for stakeholders relies on data reported on case investigation forms (CIFs) and by laboratories in order to swiftly inform the epidemiology of the disease. The incomplete collection and reporting of key variables such as symptoms, date of onset, hospitalization and travel history on CIFs as well as prolonged TAT from laboratory prevents the identification of detailed risk factors for transmission and severity of infection. The use of SORMAS as an electronic data collection tool would have improved timeliness of reporting but SORMAS was not used on real time basis as it should due to technical and connectivity difficulties and so the aim was almost defeated. There is a need to improve the electronic system and integrate it fully with the surveillance system including testing laboratories so as to facilitate timely and real time data collection and reporting.

The system had good sensitivity as it was able to detect a good number of COVID-19 cases both symptomatic and asymptomatic. The system was sensitive enough to detect a second wave of the pandemic in FCT which accounted for more than half of the entire cases recorded within the time period. However, cases of misdiagnosis due to the inability to differentiate COVID-19 signs and symptoms from that of other diseases was a major challenge, this could be attributed to the novelty of the disease and also due to the fact that many diseases share the

same combination of symptoms. The PPV of the system was low due to the non-specific nature of COVID-19 symptoms and the system did not want to miss cases, as a result, cases that did not meet the case definition were investigated and tested. The system's data quality was poor as the majority of data generated by the system were partially complete due to the absence of some key variables. This finding is in line with the report of an assessment of the national framework of COVID-19 surveillance in the United States by Ulrich et. al. [9], where incomplete and inconsistent data collection was recorded and the data was reported in a non-standardized way. However, it is in contrast to the findings by Atifa et. al. from evaluation of the COVID-19 Laboratory-Based Surveillance System in Islamabad-Pakistan where they found completeness of data and consistency of reporting to be good [6]. Surveillance systems must capture adequate and detailed information about cases to facilitate rapid epidemiological analysis of the disease and effectively inform prompt public health actions. Coordination of the pandemic in FCT needs a departure from paper-based reporting systems and move towards fully integrated electronic systems to further improve data quality. The goal of electronic reporting of IDSR data is to strengthen the disease surveillance system for prompt detection of public health events and real-time reporting, thereby enabling prompt response to outbreaks and public health actions [26, 22]. The system has good representativeness as cases were detected across all ages and geographical areas of FCT, cases were also reported from public and private healthcare facilities. Representativeness is important for planning and executing targeted interventions and monitoring progress towards containment. The stability of the system was considered to be fair as there was no steady sustainable funding from both government and partners and activities of the system had been interrupted in the past as a result of this. Nigeria's health system is irrefutably fragile as a result of incessant outbreaks that have weakened healthcare frameworks. The COVID-19 pandemic has worsened the pre-existing situation. As expected, the pandemic had overwhelming public health and socioeconomic impacts globally, resulting in a decrease in the epidemiological control of several infectious diseases [27].

It is also important to mention that the COVID-19 pandemic has uncovered Africa's inefficient and ineffective health surveillance systems. Although, Africa has recorded several outbreaks of emerging and re-emerging infectious diseases such as Ebola virus disease and other epidemic-prone diseases, less attention has been given towards surveillance system strengthening [28]. Indeed, the impact posed by the COVID-19 pandemic on health systems in the region has been catastrophic and has stressed the importance of rethinking, reflecting and focusing on lessons learned during the COVID-19 pandemic. Africa, like every other continent has been affected, and the underlying shortfalls in its health system has aggravated the situation [29]. In the wake of COVID-19 widespread, healthcare systems became overwhelmed in Africa. Short supply of skilled health workers was recorded, falling 60% below the United Nation's minimum limit, while sub-Saharan Africa has only 1%–5% of the intensive care unit beds per capita, compared to European and East Asian countries [30]. In Nigeria for example, despite the decrease in patient visits during lockdown, the increased healthcare needs for critically ill patients in inadequately equipped and understaffed intensive care units resulted in substantially less time dedicated to non-COVID-19 patient care. Nigeria's current health systems cannot efficiently cater for the increasing needs of already infected patients who require intensive care for acute respiratory diseases [31].

Infectious diseases surveillance involves continuous vigilance for health occurrences and related events to ensure quick response [22]. However, because of low engagement and actions from different stakeholders in increasing the performance strength of surveillance, COVID-19 cases may have been under-reported in Africa [31]. Although recorded cases and mortality may seem low, it has been projected that Africa will likely have one of the worst effects of the

pandemic. It is thus important to ramp up laboratory and diagnostic capacity in an effective and continuous structure across African countries for prompt detection, and accurate predictions of COVID-19 and other infectious diseases.

The inefficient surveillance strategies in Africa have led to suboptimal reporting and monitoring of diseases of public health importance, which gives a false sense of decrease in incidence rates of prevalent diseases, ultimately affecting policymaking and eradication strategies [28]. Low information-based activity from healthcare centers has led to low quality delivery of healthcare services in Nigeria. It has also affected morbidity and mortality rates, along with relevant general data on the leading infections in Nigeria [28]. The success of any health initiative is jeopardized because of low disease surveillance in Africa. This has led to disorganized healthcare sectors with poor infrastructure, low knowledge trends and low-quality healthcare delivery, which indicate low sustainability. Moreover, it has also affected the accurate assessment of the health systems as well as health promotion programs, leading to unproductive resource allocation by program investors.

It is very essential to mention that during the pandemic, several African countries around the world documented a rise in infectious diseases. COVID-19 pandemic exacerbated the spread of many infectious diseases such Dengue fever, yellow fever, measles, Lassa fever and malaria as seen in many African countries [32]. The primary prevention strategy against these diseases is vaccination and entomological control of vectors; implementation of such strategy in the continent is far below what is needed to control the diseases. The restrictions encountered due to COVID-19 pandemic led to interruption of prevention and control programs [33]. This underscores the burden and challenges of other highly infectious diseases amid COVID-19 pandemic and these diseases have impacted African countries resulting to overstraining of the already dilapidated healthcare system. For instance, there were several infectious diseases outbreak or reemergence amid the COVID-19 pandemic in Nigeria. The coexistence of other outbreaks during COVID-19 pandemic increased the burden on the country's health system mainly because the necessary response programs for these re-emerged infectious diseases were redirected to the COVID-19 national response. Particularly, during the COVID-19 pandemic, there were yellow fever, cholera and Lassa fever outbreaks in many parts of Nigeria, owing to the country's inadequate health system, which hampered the development of proper disease responses. As in other countries around the world, the burden added by COVID-19 to the public health system has led to a reduction in the epidemiological control of other infectious diseases [29]. This disappointment permitted the episodes to gain in recurrence and seriousness with sizeable mortality rates, which also included healthcare workers [34].

Africa's healthcare workforce and testing capacities are inadequate to integrate the COVID-19 pandemic and other viral infection surveillance, this had led to escalation of a chain reaction of crises in the healthcare system [28, 35]. Individuals with chronic liver disease have been added to those at risk with increased danger for critical expression of COVID-19, however, the existence of viral hepatitis, as well as malaria, tuberculosis, and HIV/AIDS does not directly escalate vulnerability in comparison to the SARS-CoV-2. The high occurrence of poor medical diagnosis among individual living with viral hepatitis in sub-Saharan Africa could be linked to the lack of SARS-CoV-2 infection restriction guidelines [36]. Late presentation, misdiagnosis or under-diagnosis of these tropical infectious diseases in Nigeria can be directly linked to the unrest in the health care delivery system during COVID-19, laboratory equipment, infrastructure and manpower have been channeled solely for the purpose of COVID-19 [36, 37].

Evaluation of COVID-19 surveillance system in FCT shows the state's adequate capacity to detect, respond and contain the disease. Morbidity and mortality data which is the main point

of measurement of COVID-19 burden, highlights the usefulness of response activities as well preventive measures taken. It was crucial to identify weaknesses and strengths of the FCT's COVID-19 surveillance system in this short period in order provide government, stakeholders and partners with robust evidence for better policymaking, strategic planning, improvement actions and sustainability of the system for monitoring these sorts of pandemics.

## Limitations

The mixed-method approach employed in this study elicited in-depth perceptions on the surveillance system attributes from respondents. However, a limitation of this study was a likelihood of bias in responses due to social desirability since the study was largely dependent on respondents' self-reporting. Moreover, the perspectives of the respondents on the surveillance system attributes was restricted to certain concepts as provided in the adapted semi-structured questionnaire. Nevertheless, a comment section was provided to enable respondents to illuminate further on the attributes. Secondly, the surveillance data retrieved and analyzed in our study could be limited due to incomplete data entry and missing variables on the electronic database. The extent of this is likely minimal since some of the missing data were keyed in manually on the excel sheet after retrieval from SORMAS. Nonetheless, the data in this study permit an early assessment of the epidemiological and clinical characteristics of COVID-19 in FCT thereby describing the usefulness of the surveillance system. The substantial proportion of missing data has prompted a systemic effort to improve data collection process and introduce electronic CIF. Lastly, we were unable to conduct KII with case managers due to restricted access at treatment centres; one-on-one interviews with these key stakeholders would have provided richer information about the system as regards management of cases. This was mitigated by using electronic surveys as an alternate method to obtain the necessary data.

## Conclusion

This study has provided an early insight into the performance of COVID-19 surveillance system in FCT, Nigeria, highlighting information necessary for health system strengthening and public health planning. Despite its many strengths, some significant weaknesses and gaps were identified in the FCT COVID surveillance system during this evaluation. These weakness needs to be addressed with a sense of urgency in order to optimize system performance. The system has played a critical role in the containment of the pandemic, demonstrated by the rapid reduction in the number of confirmed cases, well-coordinated incident management system, good case management and IPC measures. Data generated from the FCT COVID-19 surveillance system was promptly analyzed and very useful in providing information on the trend which subsequently helped in decision making, monitoring the outbreak and planning/ improvement of response activities.

The system was found to be simple, flexible, sensitive, acceptable to stakeholders, with good representativeness. However, the stability, data quality and timeliness should be improved upon. Additionally, with the high operating costs of the surveillance system and a history of dependence on external financial support, the long-term financial sustainability of the system remains uncertain. Overall, the performance of FCT COVID-19 surveillance system was rated moderate as it was observed to be addressing the public health problem for which it was instituted and also meeting its objectives to some reasonable extent although there were major issues affecting the efficiency of the system.

## Recommendations

1. The FCT Public Health Department should ensure that COVID-19 surveillance activities are fully funded, as funding from donors might not be sustainable and sufficient to cover all surveillance activities, funding was found to be a major determinant of the stability of the system.

2. Partners (WHO, AFENET, ACDC and REDISEE) should consider increasing funding and technical support made available for FCT COVID-19 surveillance at all levels.

3. The Director of Public Health should ensure that all personnel involved in surveillance activities at the state, LGA and health facility levels are trained and retrained periodically, this will improve the system's data quality and consistency in reporting.

4. The State Department of Health should provide additional logistic resources to DSNOs in rural areas, vehicles should also be made available for surveillance activities at all times.

5. The FCT Ministry of Health and FCT PHD should strengthen, expand and fully decentralize COVID-19 surveillance activities to all area councils and activate sample collection centres in selected health facilities across the area councils.

6. Constant monitoring of infection prevention and control (IPC) compliance in health facilities by FCT PHD to reduce HCW infection.

7. External quality assessment for private COVID-19 testing laboratories by MLSCN and NCDC to ensure accurate results are constantly reported.

8. Efforts to effectively improve system-wide timeliness should be directed to strengthening the first reporting stage from the testing Laboratory.

9. NCDC should make a deliberate effort to resolve the observed technical and connectivity issues associated with SORMAS so as to facilitate easy operation of the platform on real-time basis. This will resolve the issue of large amounts of backlogs of CIFs not captured on SORMAS, by so doing, the COVID-19 surveillance can then consider phasing out paper documentation, thereby achieve overall improvement of the system's timeliness and data quality.

10. The FCT PHD should improve on supportive supervision to LGAs and facilities on proper data collection, timely reporting and stock consumption reports.

11. The state ministry of health should establish more state-owned COVID-19 testing laboratories to support testing in FCT and improve result TAT.

## Supporting information

**S1 File.**
(DOCX)

## Acknowledgments

We wish to acknowledge the Director of FCT Public Health Department, Dr. Josephine Okechukwu and the FCT State DSNO Mrs. Fatima Ahmed, for their assistance, guidance and encouragement during the conceptualization of this study and facilitation of surveillance data acquisition process. We also acknowledge the FCT PHD and EOC for their immense support throughout the period of this study.

## Author Contributions

**Conceptualization:** Chikodi Modesta Umeozuru.

**Data curation:** Chikodi Modesta Umeozuru, Doris Japhet John, Lukman Ademola Lawal, Charles Chukwudi Uwazie.

**Formal analysis:** Chikodi Modesta Umeozuru.

**Methodology:** Chikodi Modesta Umeozuru, Aishat Bukola Usman, Abdulhakeem Abayomi Olorukooba.

**Project administration:** Lukman Ademola Lawal.

**Resources:** Idris Nasir Abdullahi, Muhammad Shakir Balogun.

**Supervision:** Aishat Bukola Usman, Abdulhakeem Abayomi Olorukooba, Muhammad Shakir Balogun.

**Validation:** Muhammad Shakir Balogun.

**Visualization:** Idris Nasir Abdullahi.

**Writing – original draft:** Chikodi Modesta Umeozuru, Abdulhakeem Abayomi Olorukooba, Muhammad Shakir Balogun.

**Writing – review & editing:** Chikodi Modesta Umeozuru, Aishat Bukola Usman, Abdulhakeem Abayomi Olorukooba, Idris Nasir Abdullahi, Doris Japhet John, Lukman Ademola Lawal, Charles Chukwudi Uwazie, Muhammad Shakir Balogun.

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
