## [Decision Letter · Decision Letter 0]

22 Nov 2021

PONE-D-21-30925Performance of COVID-19 Case-based Surveillance System in FCT, Nigeria, March 2020 –January 2021PLOS ONE

Dear Dr. Umeozuru,

Thank you for submitting your manuscript to PLOS ONE. After careful consideration, we feel that it has merit but does not fully meet PLOS ONE’s publication criteria as it currently stands. Therefore, we invite you to submit a revised version of the manuscript that addresses the points raised during the review process. Please submit your revised manuscript by Jan 06 2022 11:59PM. If you will need more time than this to complete your revisions, please reply to this message or contact the journal office at plosone@plos.org. Please include the following items when submitting your revised manuscript:A rebuttal letter that responds to each point raised by the academic editor and reviewer(s). You should upload this letter as a separate file labeled 'Response to Reviewers'.A marked-up copy of your manuscript that highlights changes made to the original version. You should upload this as a separate file labeled 'Revised Manuscript with Track Changes'.An unmarked version of your revised paper without tracked changes. You should upload this as a separate file labeled 'Manuscript'.

We look forward to receiving your revised manuscript.

Kind regards,

Khin Thet Wai, MBBS, MPH, MA

Academic Editor

PLOS ONE

Additional Editor Comments (if provided):

Thi is the important manuscript that focuses on the performance of COVID-19 case-based surveillance system.

Authors need to follow the suggestions of reviewers to improve the scientific rigour and readability.

Extensive language editing is deemed necessary.

Reviewers' comments:

Reviewer's Responses to Questions

**Comments to the Author**

1. Is the manuscript technically sound, and do the data support the conclusions?

Reviewer #1: No

Reviewer #2: Yes

2. Has the statistical analysis been performed appropriately and rigorously? 

Reviewer #1: Yes

Reviewer #2: Yes

3. Have the authors made all data underlying the findings in their manuscript fully available?

Reviewer #1: No

Reviewer #2: Yes

4. Is the manuscript presented in an intelligible fashion and written in standard English?

Reviewer #1: No

Reviewer #2: Yes

5. Review Comments to the Author

Reviewer #1: This paper is a good attempt to explore the performance of real time mobile and web-based COVID-19 surveillance system (SORMAS) by employing different study methods including survey, key informant interviews as well as record reviews in the Federal Capital Territory of Nigeria. While the paper reported favorable results with the surveillance system, there are several major concerns with the study.

Major comments

1.The writing is hard to follow and will benefit substantially from rigorous scientific editing.

For example, the introduction section is lengthy and two paragraphs from lines 68 to 105 could be more succinct and shorter. The paragraph from lines 106 and 119 seems to describe about the flow of the SORMAS, which could be added to explain about the study setting.

2.It is not clear which framework of performance assessment the authors used nor how it was selected and validated in their setting. Did the authors also consider assessing different dimensions of performance for mobile- and web interface?

3.Methodology could improve a lot by having a clear structure of description of each method of quantitative part (surveys and record reviews) as well as qualitative part, key informant interviews. They should include study population, key variables, tools’ reliability and validity (credibility for qualitative part), data collection (method and period) and statistical analysis.

For example, in the method section,

-what are 9 evaluated attributes of the performance assessment?

-What kind of questions did you ask to get these attributes?

-Are these attributes able to measure the outcome, performance of the surveillance system as well as user-based aspects like mobile and web interface?

-Are these tools validated? Literatures on these should be mentioned in the introduction or methods section.

4.What do the authors mean by “community case definitions” in line 210? Why are there two different case definitions? This could introduce misclassification bias.

5.In Key informant interviews, what criteria did you used to select four persons as stakeholders? Can there be other people who used the system like data entry staffs or volunteers? Are these four persons included in the quantitative survey as well?

6.Whether the surveillance system cover nationwide as well as whether this study only used data for FCT region is not very clear.

7.What do the author mean by “The system effectively detected”?

8.Why did the authors include sensitivity and PPV as performance attributes? Is sensitivity assessing the surveillance system or case definitions? If it’s case definition, it is not relevant for this study. Again, is PPV for RT-PCR or the system?

Minor comments

-Line 155: expand these “Ass. State DSNO, LGA DSNOs, SORMAS SSO,”

-COVID-19 case definitions should be mentioned in the method section before the data analysis part.

-Expand AMAC and TAT in line 226 and 227.

-Where does the figures on population size come from in Table 1?

-Mention in the method section how CFR, positivity rate and attack rate were calculated in Table 1.

-Figure titles are missing and cumbersome to figure out which one belong to which.

-Figure 1: any meaning in black and green arrow lines?

-Figure 2: it is confusing when you mentioned epidemiological week. Adding years 2020, 2021 like Figure 3 should be more clear.

-Why are there only 6 attributes in the last figure? I thought there were 9.

Reviewer #2: The manuscript entitled “Performance of COVID-19 Case-based Surveillance System in FCT, Nigeria, March 2020 –January 2021” by Umeozuru et al. conducted a cross-sectional study design, comprising a survey, key informant interview, record review and secondary data analysis to evaluate the performance of COVID-19 case-based surveillance system in FCT, Nigeria and assess its key attributes. I would like to thank the authors for this interesting work. However, there are some points that should be considered to revise the manuscript:

1. The recent COVID-19 pandemic has uncovered the region's inefficient and ineffective health surveillance systems. However, the impact posed by the COVID-19 pandemic on health systems in the region has been catastrophic, it has also stressed the importance of rethinking and focusing on lessons learned during the COVID-19 pandemic. Authors need to briefly discuss the impact of poor disease surveillance on COVID-19 response in the manuscript. Reference: https://doi.org/10.1016/j.cegh.2021.100841

2. During the pandemic, several African countries around the world have documented a rise in infectious diseases. Authors need to briefly discuss the burden and challenges of infectious diseases amid pandemic as well as discuss how the disease has impacted African countries and the strain in the healthcare system caused by COVID-19. Suggested papers: https://doi.org/10.1186/s41182-021-00356-6, https://doi.org/10.1002/jmv.27137, https://doi.org/10.1002/jmv.27169, https://doi.org/10.1002/hpm.3317, https://doi.org/10.1002/jmv.27276, https://doi.org/10.1016/j.jemep.2021.100702, https://doi.org/10.1002/hpm.3334, https://doi.org/10.1016/j.afjem.2021.10.006.

3. The authors need to include and discuss the limitations of the current study.

4. The authors need to ensure the English language is of sufficient quality to be understood. I suggest the authors to ask a colleague who is a native English speaker to review your manuscript for clarity or to use a professional language editing service where editors will improve the English to ensure that your meaning is clear and identify problems that require your review.

6. PLOS authors have the option to publish the peer review history of their article (what does this mean?). If published, this will include your full peer review and any attached files.

Reviewer #1: No

Reviewer #2: No

---

## [Author Response · Author response to Decision Letter 0]

31 Jan 2022

Comments from Reviewer 1

• Comment 1: [The writing is hard to follow and will benefit substantially from rigorous scientific editing.]

Response: Thank you for pointing this out. We agree with this comment. Therefore, we have made some reasonable scientific editing on the manuscript. We have tried to summarize the introduction section. The paragraph that describes SORMAS has also been added to explain more about the study setting, the changes can be found from lines 122 to 143 in the revised manuscript. 

• Comment 2: [It is not clear which framework of performance assessment the authors used nor how it was selected and validated in their setting. Did the authors also consider assessing different dimensions of performance for mobile- and web interface?]

Response: We are sorry that this part was not clear in the original manuscript. The surveillance system was evaluated using the “2001 United States Centers for Disease Control’s updated guidelines for Evaluating Public Health Surveillance Systems”. This evaluation was focused majorly on assessing the key attributes of the surveillance system as provided in the guideline used.

• Comment 3: [Methodology could improve a lot by having a clear structure of description of each method of quantitative part (surveys and record reviews) as well as qualitative part, key informant interviews. They should include study population, key variables, tools’ reliability and validity (credibility for qualitative part), data collection (method and period) and statistical analysis.]

Response: We agree with this and have incorporated your suggestion where applicable. The data collection sub-section under method section provided a clear description of the various methods employed in data collection, starting with key informant interview (lines 171 to 173), survey using questionnaire (lines 176 to 182), surveillance data analysis (lines 185 to 189) and record review (lines 190 to 196). We indicated the study population (lines 150 to 155, line 160 and 161) and outlined data analysis process (lines 180 to 191).

The nine system attributes assessed using the questionnaire are specified in line 180 and 181.

The questionnaire outlined various indicators/questions (range 2-5) for assessment of each system attribute. The adapted questionnaire has been provided as a supporting document.

The system attributes are able to sufficiently measure the performance of the surveillance system. The user-based aspect of the surveillance system (SORMAS) was also captured as an indicator in the questionnaire.

The tools were adapted from United States Centres for Disease Control and Prevention (CDC), 2001 Updated Guidelines for Evaluation of Public Health Surveillance Systems. The literature on this was mentioned in the method section (lines 182 to 184).

• Comment 4: [What do the authors mean by “community case definitions” in line 210? Why are there two different case definitions? This could introduce misclassification bias.]

Response: Standard and community case definitions were provided in integrated disease surveillance and response (IDSR) for COVID-19 surveillance. However, only the standard case definitions were applied for case detection and so misclassification bias is unlikely. We have revised accordingly.

• Comment 5: [In Key informant interviews, what criteria did you used to select four persons as stakeholders? Can there be other people who used the system like data entry staffs or volunteers? Are these four persons included in the quantitative survey as well?]

Response: Purposive sampling method was used to select the four key stakeholders for the key informant interview (line 175). The four stakeholders selected for the key informant interview were viewed as the major key players in the system and so are able to provide rich detailed information about system. Case managers at the treatment centers would have been involved in the key informant interview but we couldn’t engage in one-on-one interview due to heavy restriction at the treatment centers (lines 561-565). Case managers, data entry staff and some volunteers were involved in the quantitative survey.

• Comment 6: [Whether the surveillance system cover nationwide as well as whether this study only used data for FCT region is not very clear.]

Response: We are sorry that this aspect was not clear to you. This evaluation was done at the state level and covers only the Federal Capital Territory (FCT). The study population are stakeholders involved in FCT COVID-19 surveillance activities (lines 150 to 155) and the surveillance data used was that of FCT as indicated in line 172.

• Comment 7: [What do the author mean by “The system effectively detected”?]

Response: This was an attempt to describe and emphasize the usefulness of the surveillance system. We have rephrased this sentence for clarity.

• Comment 8: [Why did the authors include sensitivity and PPV as performance attributes? Is sensitivity assessing the surveillance system or case definitions? If it’s case definition, it is not relevant for this study. Again, is PPV for RT-PCR or the system?]

Response: You have raised an important point here. However, the CDC guideline utilized in this study included sensitivity and PPV as part of the nine system attributes to assess in order to fully evaluate the performance of a public health surveillance system. The sensitivity assessed the surveillance system and there are indicators in the questionnaire that were used to assess the system sensitivity (case definition is only one of the indicators). The PPV is for the system.

• Minor comment 1: [Line 155: expand these “Ass. State DSNO, LGA DSNOs, SORMAS SSO,”]

• Response: We have effected this correction.

• Minor comment 2: [COVID-19 case definitions should be mentioned in the method section before the data analysis part.]

Response: This correction has been made

• Minor comment 3: [Expand AMAC and TAT in line 226 and 227.]

Response: This has been done

• Minor comment 4: [Where does the figures on population size come from in Table 1?]

Response: A reference has been provided for the figures on population size (reference number 20)

• Minor comment 5: [Mention in the method section how CFR, positivity rate and attack rate were calculated in Table 1.]

• Response: This has been done (line 200 to 205)

• Minor comment 6: [Figure titles are missing and cumbersome to figure out which one belong to which.]

Response: This has been corrected.

• Minor comment 7: [ Figure 1: any meaning in black and green arrow lines?]

Response: The black describes the reporting flow (feedforward) while the green arrow describes the feedback flow. Surveillance information flows from the lower to higher levels for onward public health actions based on the final laboratory outcome, while feedback goes in reverse direction as shown in figure 1.

• Minor comment 8: [Figure 2: it is confusing when you mentioned epidemiological week. Adding years 2020, 2021 like Figure 3 should be more clear.]

Response: This correction has been effected

• Minor comment 8: [Why are there only 6 attributes in the last figure? I thought there were 9.]

Response: Figure 6 has been updated to include 9 attributes

Comments from Reviewer 2

• Comment 1: [The recent COVID-19 pandemic has uncovered the region's inefficient and ineffective health surveillance systems. However, the impact posed by the COVID-19 pandemic on health systems in the region has been catastrophic, it has also stressed the importance of rethinking and focusing on lessons learned during the COVID-19 pandemic. Authors need to briefly discuss the impact of poor disease surveillance on COVID-19 response in the manuscript.]

Response: Thank you for pointing this out, we have made the additions from line 484 to 517

• Comment 2: [During the pandemic, several African countries around the world have documented a rise in infectious diseases. Authors need to briefly discuss the burden and challenges of infectious diseases amid pandemic as well as discuss how the disease has impacted African countries and the strain in the healthcare system caused by COVID-19.]

Response: We appreciate this input. We have revised accordingly from line 520 to 530.

• Comment 3: [The authors need to include and discuss the limitations of the current study.]

Response: We agree with this and have incorporated a section for study limitation from line 558 to 575.

• Comment 4: [The authors need to ensure the English language is of sufficient quality to be understood. I suggest the authors to ask a colleague who is a native English speaker to review your manuscript for clarity or to use a professional language editing service where editors will improve the English to ensure that your meaning is clear and identify problems that require your review.]

Response: We totally agree with this. We have made efforts to improve English language in our manuscript and have corrected some grammatical errors.

---

## [Decision Letter · Decision Letter 1]

18 Feb 2022

Performance of COVID-19 case-based surveillance system in FCT, Nigeria, March 2020 –January 2021

PONE-D-21-30925R1

Dear Dr. Umeozuru,

We’re pleased to inform you that your manuscript has been judged scientifically suitable for publication and will be formally accepted for publication once it meets all outstanding technical requirements.

Kind regards,

Khin Thet Wai, MBBS, MPH, MA

Academic Editor

PLOS ONE

Additional Editor Comments (optional):

Reviewers' comments:

Reviewer's Responses to Questions

**Comments to the Author**

1. If the authors have adequately addressed your comments raised in a previous round of review and you feel that this manuscript is now acceptable for publication, you may indicate that here to bypass the “Comments to the Author” section, enter your conflict of interest statement in the “Confidential to Editor” section, and submit your "Accept" recommendation.

Reviewer #1: All comments have been addressed

Reviewer #2: All comments have been addressed

2. Is the manuscript technically sound, and do the data support the conclusions?

Reviewer #1: Yes

Reviewer #2: Yes

3. Has the statistical analysis been performed appropriately and rigorously? 

Reviewer #1: Yes

Reviewer #2: Yes

4. Have the authors made all data underlying the findings in their manuscript fully available?

Reviewer #1: Yes

Reviewer #2: Yes

5. Is the manuscript presented in an intelligible fashion and written in standard English?

Reviewer #1: Yes

Reviewer #2: Yes

6. Review Comments to the Author

Reviewer #1: (No Response)

Reviewer #2: I think that this paper is vastly improved and thank the authors for putting good efforts to implement all of my suggestions. Satisfied with the response. Recommended for publication.

7. PLOS authors have the option to publish the peer review history of their article (what does this mean?). If published, this will include your full peer review and any attached files.

Reviewer #1: No

Reviewer #2: No

---

## [Editor Report · Acceptance letter]

5 Apr 2022

PONE-D-21-30925R1 

Performance of COVID-19 case-based surveillance system in FCT, Nigeria, March 2020 –January 2021 

Dear Dr. Umeozuru:

I'm pleased to inform you that your manuscript has been deemed suitable for publication in PLOS ONE. Congratulations! Your manuscript is now with our production department. 

Kind regards, 

on behalf of

Dr. Khin Thet Wai 

Academic Editor

PLOS ONE